# Administrators and Students on E-Learning: The Benefits and Impacts of Proper Implementation in Nigeria

**Esen Sucuoğlu and Azubike Umunze Andrew ***

Department of Educational Administration and supervision, Near East University, Nicosia 99138, Cyprus; esen.sucuoglu@neu.edu.tr
* Correspondence: princeazubike82@gmail.com

**Abstract:** The quest for better education and knowledge acquisition has triggered the introduction, acceptance and incorporation of e-learning into Nigerian learning. The introduction of the concept of e-learning to Nigerian learning can be dated back to the 1980s, when reputable Nigerians enrolled in several universities in London. In addition, the introduction of e-learning to a premier university in Nigeria, rooted in the college of Ibadan, led to greater interest, causing locals to seek extramural work and other studies at Oxford University. This study examines the impacts that proper educational administration, policy making and implementation, as well as the adoption of e-learning, can have to fix the dilapidated Nigerian educational structure. A quantitative method of data collection was used, through well-structured questionnaires for both administrators and students issued to the four universities sampled in this study. A total of 240 questionnaires were issued to respondents, with 60 each to the different universities and with 30 each for both students and administrators. A total of 180 were retrieved, and descriptive analysis was carried out with SPSS (23). Internal consistency was determined with Cronbach's alpha, having an internal consistency of 0.78. The findings show that all the administrators were graduates with a minimum of a Bachelor's degree. It was revealed that 32 (17.8%) of the students possessed smartphones as gadgets for e-learning and that administrators contributed to the enhancement of student performance, hence creating impacts in their examination grades, with a mean of 2.66, being rated 'Good' for their performance. Unfavorable government policies and unprofessionalism of administrators in e-learning implementations were the major constraints, with a mean of 4.6. The cost of the procurement of the needed resources (data) for e-learning also impacts e-learning. Internet resources used by the students contributed to huge success in e-learning for 28 (24.6%) and 24 (21.9%) students. Although the constraints limit the effectiveness of e-learning in Nigeria, it also impacts student advancement compared with the face-to-face learning process. The government's proactive measures will improve e-learning.

**Keywords:** e-learning; administrators; proper implementations; well-structured questionnaires; un-professionalism

## 1. Introduction

Education is an investment that needs to be made by every nation and all people aspiring for rapid and robust growth in their economic, social and even political aspects of the nation's needs, in order for them to attain the height and level of productivity they need [1]. In recent times, the quest for more advanced security and a more positively impacted educational system, capable of providing solutions to learning and teaching environments, has been a problem in the field of education. Several countries have sourced means to resolve this issue but with little or no achievement made in the past [2].

Schools across the globe have been shut down amidst the emergence of the COVID-19 pandemic, which originated from Wuhan, China, thus causing the authorities of the world to initiate lockdown as an emergency measure to manage the spread of COVID-19, including the closure of schools [3]. This development has brought about the acquirement

and implementation of e-learning in most universities [4]. The cost of obtaining required instruments has allowed the continuation of learning and teaching and has also allowed for advancement from normal face-to-face learning patterns, with excellent results and grades [5]. The success of e-learning in the COVID-19 era has had to deal with the satisfaction that e-learning has brought to the end users and with the system that e-learning has to offer [6,7].

The Nigerian educational system has observed this, and it has arranged for the implementation and proper administration of e-learning [8].

E-learning is relevant for any country that seeks positive impacts in the educational sector and hence is widely used in this sector, yielding adequate and appropriate benefits [9].

Administration and administrators in schools and in educational spheres deal with different social processes that allow the people or administrators to identify, maintain, control and stimulate both formal and informal human resources within a given and well-organized system, such as schools [10,11]. The administration of good e-learning is entrusted into the hands of people at different apex positions in the educational sector, which spawn from the teachers, head of departments, deans and council members who make sure that the administration and implementation of e-learning are achieved along with the desired results [12].

In other to be a good teacher or administrator who is able to impact a learner's interest, an individual needs to possess some attributes and unique features which involve the internet and multimedia in order to effectively pass the information they want to give to timid students. These resources are readily available but require proper utilization for optimal output [13,14]. The implementation of e-learning across different academic levels, from the primary, secondary and tertiary institutions, is accompanied by several gaps which need to be addressed, and these gaps require the attention of both new and old schools for proper administration to be achieved [12]. The Nigerian educational system is advancing in its quest for well-organized and standard educational service delivery to help students make use of the resources which are readily available to them, hence the need for good administration and implementation of e-learning facilities in most academic spheres [15]. This administration and implementation require the principal and concerned members (the Head of Departments, Deans of Faculties and the council members of the universities) to come together and adopt the latest internet, technological, multimedia and other relevant tools that make learning possible and that allow the desired aim to be achieved. [16], highlighted e-learning to be a learning procedure that deals with the learning and use of computers. This is a result of the age of greater advances in the learning and education sector. E-learning entails the process of creating an enabling environment for students or their employers to be effectively trained in the process of continuous learning and teaching [17].

E-learning provides several opportunities such as financial benefits mostly in cases where there is lesser or limited provision of the required basic amenities. The need to address the task is being set up, thereby allowing the utilization of a cyber café in achieving the target, hence generating income and financial assistance [18].

The effectiveness of e-learning is solely dependent on the aggregation of more relevant and important technologies that play vital roles in making sure that the use of e-learning is easy and simple to understand [19].

An enabling environment plays a vital role in the acceptance, implementation and actualization of e-learning in the educational sector in order to drive and achieve the success that it has to offer [20]. The use of technologies such as multimedia, with the most important being informational tools, makes it possible to achieve flexibility in the understanding and usage of e-learning facilities in schools and institutions of higher learning, which is beneficial to both administrators and students [21,22]. Administrators of the Nigerian educational system have identified e-learning as being advantageous to both students and administrators, hence advocating for good quality and relevant gadgets to further enhance

learning benefits for students [23]. The implementation of e-learning in the Nigerian educational system, compared with other developed nations, has suffered major setbacks with respect to meeting standards and acquiring needed materials and manpower, and the provision of basic amenities, such as light, gadgets, ICT, etc., could not be met [24].

The educational sector in Nigeria is faced with several challenges amid the novel COVID-19 pandemic, which has ravaged the educational sector as well as other financial institutions [25].

According to [26], the utilization of e-learning is accompanied firstly by its adoption before students can use it in both private HEIs and in public schools.

## 2. Literature Review

### 2.1. Concept of School Management

The optimal administration and incorporation of the best trends in educational sectors provide a good chance of a higher performance in schools, including either universities or secondary schools, where they are implemented [27]. The introduction of e-learning to a premier university in Nigeria, which has its roots in the college of Ibadan, has led to greater interest, which has caused locals to seek extramural work and several other studies at Oxford University [28].

The smooth running of education and e-learning in the Nigerian educational sector is hampered by several limitations and setbacks, including the curriculum, which is not adequately prepared, the lack of adequate electricity, the unprofessionalism of the staff and other factors [29,30].

Ref. [31], in his view, asserted that education needs resources and proper mobilization by the parents and guardians of the students in order to meet demands and what is expected of schools regarding academic success. It is a process of mobilizing school resources toward the achievement of desirable educational goals. School administration is a process of activation that requires expertise and training in educational principles and practices in order to ensure proper management to achieve results in education [11].

### 2.2. Concept of E-Learning

E-learning, also known as "electronic learning", is associated with the use of internet resources and other such tools to gain knowledge, with the maximum utilization of multimedia, technologies and ICT in making sure that the desired aim is achieved [9]. It is a system that helps in the effective passing and receiving of information from one source to another, i.e., from teachers to students [9,32].

According to [18], e-learning is an educational method for which popularity is on the rise in all aspects and endeavors of life, and as such, it has gained a high level of recognition and acceptability in the 21st century. In most formal school systems, however, e-learning is faced with many expectations, such as conforming to educational curriculum. E-learning entails the process of creating an enabling environment for the students or their employers to be effectively trained in the process of continuing to teach [17]. Facilitators and instructors of e-learning make learning more interesting and create more benefits; therefore, further exposure is seen in the learning process [33].

Ref. [2] acknowledged that, following many advances that have been observed, the rapid change in technological progress and the most recent globalization trend in the quest for higher educational performance, as well as the elimination of boundaries among students, have led to the adoption of new methods and perspectives in educational practice, such as e-learning. According to the Federal Republic of Nigeria [34], formal education in Nigeria is controlled by a series of policies.

### 2.3. Benefits of E-learning in the Educational Sector

One of the benefits of e-learning is the fact that e-learning saves 50% of time and 40–60% of costs when used well [35]. It also provides the following:

- **Location flexibility:** E-learning provides learners with flexibility, and the same is also provided for teachers. This makes learning easier for both parties notwithstanding the location.
- **Good access to instructors:** Learners can acquire a better understanding of the concepts being taught, as the materials that are provided can be re-accessed along with knowledge from instructors.
- **Reliable learning skills:** Certain circumstances have been observed by researchers to be of great benefit to learners using online learning, compared with the classroom learning process. Improvements in reliable learning skills are seen online, compared with normal face-to-face learning that is adopted in most countries' educational systems [35].

With the inception of e-learning, there are no barriers, as online learning provides a global perspective because it connects users beyond the boundaries of their location.

### 2.4. E-Learning in Nigerian Universities

Universities or higher institutions are seen as well-organized educational settings above the secondary level of learning, which offer degrees, diplomas or other relevant certificates that are widely recognized and accepted [28]. Higher institutions provide humans with opportunities to run the economy, which hence provides the economy with greater transformation in the society in which it is practiced [31]. [36] posited that e-learning has, in recent times, brought about a change in the relationship between teachers and students with respect to learning, which has transformed the learning process from the traditional education system. The benefits of implementing e-learning in the country are necessitated by the government through the Federal Ministry of Education to initiate policies that enable growth and stability but that are yet to be fully implemented for optimal performance [37]. Despite the educational benefits of e-learning, there is also a financial benefit, but this is always taken for granted by both students and administrators [18]. Continual advancements in information technology have provided the educational sector with the necessity to advance their teaching, learning and research methods, since these proactive measures are needed to effectively perform the duties assigned to them for optimization and for running smoothly [9,38].

### 2.5. Administrators of E-Learning in Nigerian Universities

Different principal officers play key roles in the administration and implementation of policies in organizations where they serve and hold principal offices. The effects are widely felt in the educational aspect, where administrators seek the most suitable methods of improving the educational performance of their students, mostly in higher institutions of learning [15]. The administrators of universities in Nigeria have roles to play in ensuring that e-learning and other relevant teaching strategies are for the benefit of the students [39].

AFRIHUB, which is responsible for providing ICTC in most Nigerian universities, agrees with Mr. Chukwuemerie Nnamdi that over (18) tertiary institutions in Nigeria utilize the help and assistance from them to unleash the "power of ICTs in almost all Nigeria and other African countries for the development of human capacity building and also to enhance economic empowerment, not only in the educational sectors" [40]. These tertiary institutions where programs are currently in operation are the Michael Okpara University of Agriculture, also known as Umudike; the Federal University of Technology Owerri (FUTO); Nnamdi Azikiwe University; the Awka University of Nigeria Nsukka:- and Enugu Campuses (UNN, UNEC); the University of Abuja; the University of Benin; Minna (FUTA); the Federal College of Education, Technical, Omoku; and the University of Calabar, amongst others in the country. With these establishments, plans are also in preparation to start ICT operations in more tertiary institutions in the country under a public–private partnership (PPP) [40]. The administrators of different universities in the country have started putting things in place to help standardize the internet and e-learning

programs, which are seen to be of great importance to the academic environment and to the economy of the nation as a whole.

According to [12] in their study of the role of educational administrators in the implementation e-learning programs, carried out a study focused on the challenges faced by education administrators in e-learning implementation and on ways to remedy them. The study was implemented using mixed research methods (pragmatism research philosophy) for data collection, and analysis was conducted by [41], with primary administrators and school heads (principals used in the study) using well-structured questionnaires and with random data collection techniques. The findings of the research show that most of the school heads are degree holders and inspectors with a Master's degree in educational administration, but none of them had a relevant degree equivalent in the ICTC spheres to help them be more effective in teaching and e-learning implementation [27]. The study concluded that no optimal progress can be achieved in e-learning in schools without well-trained ICTC personnel being brought into the educational sector of Zimbabwe, despite the government's efforts to avert the problems faced.

### 3. Methodology

**Research Design:** A survey research design was adopted that uses a randomized standardized sampling technique, with valid, well-structured questionnaires aimed at considering each respondent's attitude towards and perception of e-learning. A quantitative method was used for data collection from administrators and students.

**Instrumentation and data collecting tools:** The collection of data was carried out by administering 240 well-structured questionnaires that addressed the problems of interest to the participants, who were both students and administrators. A total of 60 questionnaires were administered in different universities, with 30 each for both students and administrators. This data collecting tool was administered to both the administrators and the students to collect their ideas according to the information provided in the questionnaire. Randomized sampling and quantitative data methods were used to gather the opinions of administrators on their understanding and perception of e-learning and on the benefits that the learning procedure can offer to them.

**Method of questionnaire administration:** The data collecting tool (questionnaire) with a closed-ended pattern of questions to address the situation was administered by the researcher and through self-administration by the respondents. The paper and pen method was used, hence offering the respondents the same pattern of questions and allowing the easy retrieval of the filled questionnaires.

**Determining a student's performance:** The performances of students were determined based on overall performances put in during examinations, and different grade levels were designated as Poor, Average, Good, Very Good, etc. These performances of students were provided by the administrators and other staff who assumed roles in different universities, with those who used these grade levels as criteria for the performances of students.

**Subject Sampling:** The study was carried out in the southern part of Nigeria, and four public higher institutions were included, which were carefully selected from three states (Rivers State, Bayelsa State and Akwa-Ibom State), with two universities being from Rivers State (University of Port Harcourt and National Open University), the University of Uyo being from Akwa Ibom State and the Federal University Otueke being from Bayelsa State. The coordinates of the University of Port Harcourt are located at longitude 4°90′69″ N and latitude 6°91′70″ E; for the National Open University (NOUN) Port Harcourt Study Center, the coordinates are 4°86′81″ N and 6°95′66″ E; for the University of Uyo, the coordinates are 5°04′08″ N and 7°91′98″ E; and for the Federal University Otueke, the coordinates are 4°47′27″ N, and 7°91′98″ E.

**Data analysis procedure:** The data were retrieved from both student and administrator respondents regarding e-learning in the various universities. The administrators were the deans of faculties and the heads of departments of the universities. The data

were coded into Statistical package for Social Sciences SPSS Statistics 23.0, and a statistical analysis was performed using descriptive statistics and frequencies, with a significance level of 0.5. The reliability of the instrument was determined and tested using Cronbach's alpha ($\alpha$) formula: $\alpha = K/(K-1) [1 - \Sigma (S2\ y)/(S2\ x)]$ [42]

where K is the $n$ = Number of test items;
$\Sigma S2\ y$ = is the sum of item variance;
and S2 x = is the variance of the total score.

## 4. Results

*Findings of the Study Based on Research Questions*

*Research Question 1:* Who are the principal administrators, technical personnel and others that facilitate e-learning in tertiary institutions, with impacts on student performance?

Table 1 below gives a summary of the educational qualifications of the administrators and respondents from the different universities used in the study. It can be observed from the table that, out of the 180 respondents, all of the administrators were graduates with relevant qualifications, which offered them the privilege of impacting students and being able to effectively utilize e-learning strategies in teaching students. The lecturers had all obtained degrees, from B.Sc./B.Ed. degrees held by 16 (8.9%) lecturers to Ph.D. degrees held by 48 (26.7%) lecturers. Additionally, the heads of departments (HOD) who responded had a frequency of 8 (4.4%) and 12 (6.7%), and the deans of different faculties that engaged in the study had frequencies of 39 (21.7%) and 21 (11.6%) for B.Ed., PGD and Ph.D. degrees, educational background of the administrators are shown in Figure 1 below.

**Table 1.** Principal administrators and technical staff that impact e-learning in the tertiary institutions used in the study.

| S/N | Degree | Position | No. of Respondents | Percentage (%) |
|---|---|---|---|---|
| 1 | B.Sc./B.Ed. | Technicians | 16 | 8.9 |
| 2 | M.Sc./M.Ed. | Lecturer | 36 | 20.0 |
| 3 | Ph.D. | Lecturer | 48 | 26.7 |
| 1 | B.Sc./B.Ed., M.Ed. | HOD | 8 | 4.4 |
| 2 | B.Sc./B.Ed., M.Ed. and Ph.D. | HOD | 12 | 6.7 |
| 1 | B.Sc./B.Ed., M.Ed. and Ph.D. | Dean | 39 | 21.7 |
| 2 | B.Sc., PGD, M.Ed. and Ph.D. | Dean and other council members | 21 | 11.6 |
| | **Total** | | **180** | **100%** |

*Research Question 2:* What tools and strategies are required to necessitate effective e-learning in tertiary institutions in Nigeria?

Table 2 below gives an overview of the tools and strategies that are employed by the administrators of different universities in Nigeria in order to achieve smooth running of e-learning and in order for students to make the best of it. The different gadgets and tools used by these students range from gadgets in their possession, such as smartphones and personal computers (PCs), to projectors and other tools used. Additionally, the responses from the students indicate that 32 (17.8%) of the respondents agreed that smartphones play a role in education and are predominantly used, whereas 9 (5%) disagreed with that. With regard to PCs and desktops, 17 (9.4%) and 25(13.9%) agreed and disagreed, respectively, with the availability of resources and tools and the use of them; 8 (4.4%) and 35 (19.4%) agreed and disagreed, respectively, with projector usage; and 22 (12.2%) and 32 (17.8%) agreed and disagreed, respectively, with the presence of functional e-library accessibility to students.

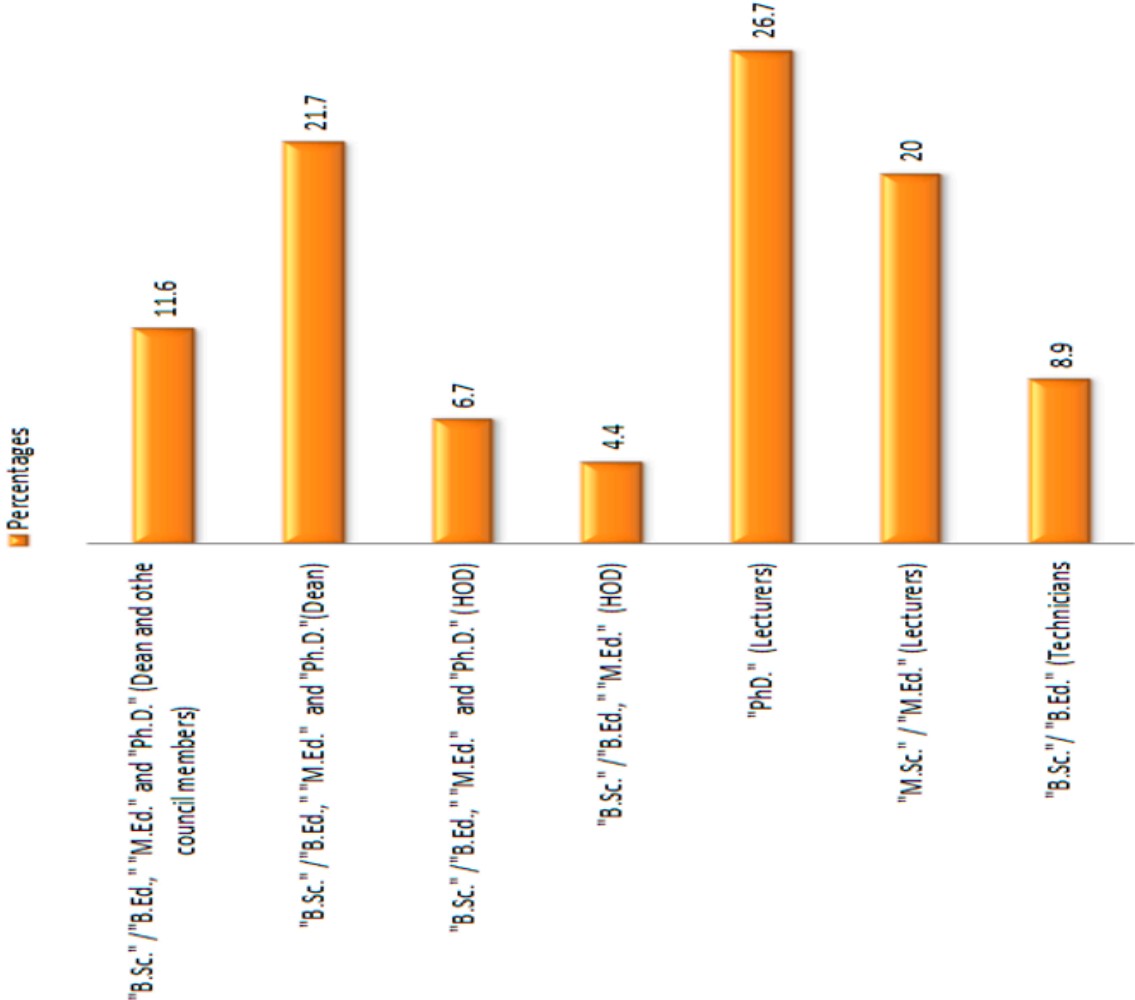

**Figure 1.** Educational background of administrators and other technical personnel.

**Table 2.** Tools and strategies for effective e-learning Implementation.

| S/N | Tools Used, Strategies and Effectiveness | Responses | | Percentage (%) | |
|---|---|---|---|---|---|
| | | Yes | No | Yes | No |
| 1 | The use of smart and Android phones in learning | 32 | 9 | 17.8 | 5.0 |
| 2 | Personal computers and desktops are made available for teaching and learning | 17 | 25 | 9.4 | 13.9 |
| 3 | Projectors are used in learning | 8 | 35 | 4.4 | 19.4 |
| 4 | Functional e-library and open access for the students for learning purposes | 22 | 32 | 12.2 | 17.8 |
| | **Total** | **180** | | **43.8%** | **56.1%** |

*Research Question 3:* Is there any relationship between the administrators of e-learning implementation and the performance level of students compared with face-to-face learning?

Table 3 below shows that the way situations are handled and the overall contributions of administrators towards the success of e-learning in Nigerian universities have an impact on the performances of students, either in a positive or in a negative way. These impacts give a complete explanation of the way forward, and headway has been observed in the educational sector. The study shows that the formation of good policies by administrators has a positive impact (2.86) on the output of students during e-learning; allocating tasks

to be carried out by students has an impact (2.80); student perceptions of the openness of administrators to them is a factor that has not strongly impacted them positively (2.26); and frequent references to online resources by administrators to the students significantly impact them and their performances (2.72). Additionally, the performance rates of students were graded as being poor, good and very good.

**Table 3.** The mean performance and remarks on the contributions of administrators towards performance compared with face-to-face learning.

| S/N | Contributions of Administrators Toward E-Learning Implementation for Students, with Regard to Student Performance with E-Learning Compared with Face-To-Face Learning | Scores with Attitude Testing | Mean Performance of Students | Remarks |
|---|---|---|---|---|
| 1 | Formulation of relevant and positive policies to aid e-learning for students and lecturers. | 60–90% | 2.86 | Very Good |
| 2 | Constant allocation of tasks to students to boost their overall performance. | 60–80% | 2.80 | Very Good |
| 3 | Being open to students expressing the challenges they are facing. | 20–40% | 2.26 | Poor |
| 4 | Frequent references to online resources for effective e-learning and optimum performance. | 35–50% | 2.72 | Good |
| | **Grand Mean** | | **2.66** | |

*Research Question 4:* What impact has the implementation of e-learning had on the receivers and providers, i.e., students and administrators?

From Table 4 below, it can be seen that the impacts of e-learning on both students and administrators are enormous. Moreover, as they cannot be ignored, the assessment of the impacts of e-learning on both students and teachers or administrators was examined and rated below a <50% and >70% scale, respectively, and the percentages of the outcomes based on the respondents were determined.

**Table 4.** Frequencies and percentages of the tools and strategies for effective e-learning implementation.

| S/N | Impacts of E-learning on Student Performance | Frequency <50 | (%) | Frequency >70 | (%) |
|---|---|---|---|---|---|
| 1 | Examination grades of e-learning compared with face-to-face learning | 11 | 16.7 | 28 | 24.6 |
| 2 | Impacts on exposure to and effective use of computer resources | 9 | 13.6 | 19 | 16.7 |
| 3 | Impact on the rate of examination malpractice | 14 | 21.2 | 24 | 21.0 |
| 4 | Impacts when compared with face-to-face learning | 7 | 10.6 | 12 | 10.5 |
| 5 | Contribution to the exposure of knowledge and self-development of e-learning compared with face-to-face learning. | 25 | 37.9 | 31 | 27.2 |
| | **Total** | **66** | **100%** | **114** | **100%** |

Thus, most aspects of the learning and expression of understanding with respect to success, examinational misconduct, comparison with face-to-face learning and cost-effectiveness were examined. The results show that the success ratio was high, according to 28 (24.6%) participants, and that some factors that bring about optimal performance had a positive impact (19 responses (16.7%)). The impacts of e-learning on examination malpractice and misconduct were lower (14 responses (21.2%)). The impacts of e-learning compared with face-to-face learning in a university show a frequency of 7 responses (10.6%), and the cost of engaging with and implementing e-learning for the benefit of the students had little impact, with a frequency of 25 responses (37.9%).

## 5. Discussion of Findings

### 5.1. Educational Background of Administrators

The demographic and educational status of the administrators that participated in the study show that all of the administrators had passed through one level of education or

another before they attained an administrative rank in their respective universities, and they had obtained a minimum of a B.Sc. and B.Ed. in their respective fields of study, potentially up to the Ph.D. level. The findings reveal that 26.7% of the administrators had a Ph.D. degree, and 21.7% also possessed a B.Sc./Ph.D in order for them to attain the position of a dean in their respective schools, thus having been exposed to the importance of e-learning. They should, therefore, be able to teach their students about the necessity of implementing this study method in Nigerian educational systems. This study has some similarities with the research of [14], who recorded that 35% of teachers and administrators held a Ph.D.

*5.2. Needed Gadgets for E-Learning*

For optimal impacts on the e-learning system, certain tools and different materials are needed for reflective and impactful learning, but these are not always available to students, except for smartphones, which serve as major tools, according to 32 (17.8%) of the participants. Additionally, other tools such as projectors are not often used for clearer explanation and for better understanding of the subject matter, thus making the identification of the process of ideas seem more complicated. These findings are similar to those of [39], who identified that CD-ROMs, the internet and other computer software packages are gadgets needed for e-learning to be effective. E-learning requires functional ICT and the accessibility of e-libraries for students, which are not always available to them, thus making it hard for students who do not have smartphones to have access to the provided materials for the learning process [37].

*5.3. Contributions of Administrators*

With regard to the contributions of administrators in the implementation of e-learning, and for them to ensure that the system succeeds despite facing some challenges, much is expected of them regarding resolving problems. Apart from them having an integral role to play in administration and implementation, they are placed in the position of counseling students in order to help build the right approach, and this is achieved by the formulation of policies, giving tasks to students, encouraging them to utilize internet resources for more understanding, etc. The findings from this study (2.66) are similar to those of [8], who observed a mean total of 2.77, necessitating the need for administrators to be active in proper e-learning implementation.

This research has shown many limitations, and the effective implementation of e-learning has suffered some setbacks, needing not only adjustment, but also an overhaul of the whole system, as the identified factors and limitations are faced by everyone in the nation; as such, they affect all sectors. The limitations include the lack of constant power supply, unprofessional attitudes of administrators, the lack of needed facilities to enhance e-learning and to achieve optimal performance and insufficient funds to invest in the educational sector.

This research would have adopted both the qualitative method and direct face-to-face data collection, but due to the COVID-19 situation, it was limited to the quantitative method, which concerns people's opinions and thoughts about the concept of e-learning in Nigeria's educational system.

Further research to address the hindrances associated with e-learning in Nigeria's educational system is worth looking at. The current research seeks to provide solutions to the impacts of e-learning and to expand upon the benefits, but little or no impact has been made regarding the factors that cause deficiency in e-learning; hence, further research needs to be carried out on factors that negatively impact e-learning in Nigerian educational sectors.

## 6. Conclusions

The impacts of e-learning on the performance of students work in line with the efforts of administrators and technical staff in universities who make sure that proper implemen-

tation is carried out for students to achieve optimal performance in the learning procedure, which, overall, means that it differs from face-face learning. This study, which focuses on e-learning and the key players who influence and implement them, has shown great advancement and impartation on students with respect to good grades, exposure to several and relevant technological skills, etc. The electronic learning process in other Nigerian educational systems will be of immense and great benefit to the drowning educational system that has been practiced in the nation in the past.

*Recommendations for Further Study*

The need to understand various roles played by administrators in ensuring electronic learning is achieved; however, the current study is centered on students and the benefits they derive from e-learning through their attitudes. Hence, more detailed studies that show the strategies which administrators use for effective learning and teaching strategies should be incorporated and studied.

**Author Contributions:** Conceptualization, E.S. and A.U.A.; methodology, E.S. and A.U.A.; software, E.S. and A.U.A.; validation, E.S. and A.U.A.; formal analysis, E.S. and A.U.A.; investigation, E.S. and A.U.A.; resources, E.S. and A.U.A.; data curation, E.S. and A.U.A.; writing—original draft preparation, E.S. and A.U.A.; writing—review and editing, E.S. and A.U.A.; visualization, E.S. and A.U.A.; supervision, E.S. and A.U.A.; project administration, E.S. and A.U.A.; funding acquisition, E.S. and A.U.A. All authors have read and agreed to the published version of the manuscript.

**Funding:** The research was fully funded by me the researcher and also the co-researcher also, advises, corrections and instructions were some of the contributions made by the co-researcher of the research work.

**Institutional Review Board Statement:** The study was conducted in accordance with the Declaration of Helsinki, and approved by the Institutional Ethics Committee of NEAR EAST UNIVERSITY, CYPRUS.

**Informed Consent Statement:** The concent of the respondents was been sought for in cause of the study with written statement on the data collection tools (Questionnaire) stating clearly that the data was for the sake of the study and that their data will be handled with confidentially.

**Data Availability Statement:** The data used in the research could only be released on request from corresponding authors (the reason for this was the ethical guidelines that was employed while taking the data with promise of a confidential handling of the data).

**Conflicts of Interest:** The authors declare no conflict of interest.

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
