# Peer review of "Administrators and Students on E-Learning: The Benefits and Impacts of Proper Implementation in Nigeria"

_electronics, doi:10.3390/electronics11101650_

Round 1

Reviewer 1 Report

Manuscript is focused on e-Learning application and the impact of all involved parties and the surrounding circumstances that affect the success of e-Learning at Universities.

This paper is submitted to the Electronics journal, section: Computer Science & Engineering, special Issue: "The Effects of the COVID-19 Pandemic on the Digital Competence of Educators". Currently, I do not see in this paper any relation of the manuscript content regarding COVID-19 Pandemic and the influence on Digital Competence of Educators. 

Title of the manuscript is confusing - how possibly could administrators (i.e. administrative or technical personell at university) be engaged as digital educators? Administrators could influence the e-Learning outcome with their technical or administrative support, but they are not digital educators! It is stated that "The administrators being the Deans of faculties, Head of Departments and Lecturers of the Universities." Lecturers are not administrators. Title could be changed to present the essence of this paper, which is properly said in the last paragraph of the introduction: "Enabling environment plays key and vital role in the acceptance, implementation and 
actualization of e-learning in the educational sector".

Authors are affiliated with Cyprus institution and they have a manuscript related to the Nigeria. It is very unusual to have such a manuscript and I belive it should be explained in the manuscript introduction how the researchers from Cyprus have conducted research in Nigeria.

The topic is not novel. In fact, text related to application of IT and e-Learning in Nigeria is somehow familiar to me. I could not state that this paper is completely authentic or plagiarism, but it seems to me that read similar paper before. It is very important for authors to provide literature review related work regarding similar research conducted in Nigeria regarding the e-Learning application at Universities. In fact, there is no Literature review section, that should be after introduction and before methodology section.

The research is based on questionnaires (with appropriate sample size), but, in fact, questionnaires represent attitudes of people included in the research sample. Statistics is performed with basic percentage and mean value, while the use of SPSS tool has been stated. 

Formulation of some sentences, including title, are not written in appropriate English writing style and with appropriate sentence construction, which leads to confusion in reading the text. Sentences are sometimes too long, such as in conclusion. Sometimes, the "," symbol is ommited, when necessary, but also sometimes it is put at the wrong place. The English grammar and writing style should be improved carefully in the whole paper.

Conclusion of the paper is poor. The conclusions are formulated as well-known statements, such as "government in ensuring that the implementation and utilization are fully utilized
has seen several delimitations and faced with a lot of challenges and weaknesses as well". There are no significant results emphasized - to be provided in conclusion, comparing to previously published papers. There is no future research directions in the conclusion section. 

Author Response

(1). The topic of the research still remains the same "Administrators and Student on E-Learning, the Benefits and Impacts of Proper Implementation in Nigeria"

(2) Regarding both authors been affiliated with Cyprus University: The research was conducted in Nigeria by the corresponding author (Azubike Umunze Andrew) who happen to be a graduate student of the university but for the sake of the research went back to Nigeria (Southern Part) to carry out the research. And with the help of the supervisor (Esen SucuoÄŸlu) Who is a Professor in Cyprus University.

(3) About the Questionnaire: Yes Indeed Questionnaire (Quantitative Method of data collection) is about the attitude of people, it was selected as the best tool since the implementation of e-learning in Nigeria is still not accepted by all, hence the schools that has accepted it is necessary to get their view regarding e-learning hence the Questionnaire.  And the use of SPSS was to determine the mean, percentages etc.

(4) Formation of sentence and Conclusion: The grammar has been checked and the conclusion also has been updated. the corrections to this effect has been highlighted in reds to show the changes and adjustments been made to the work as regards your review.

Thank you so much.

Reviewer 2 Report

Through this article, an attempt is made to outline the benefits and effects of digital education with distance learning and to capture the experiences of both educators and students involved. The introduction to this article is quite short and does not adequately describe the topic. In addition, there is no literature review section in this article and the methodology is not clear enough. The reference section needs to be updated with more recent articles (2018-2021). It will be helpful to include in the point of the methodology a flowchart that describes in detail the research design that was followed. The discussion section does not mention extensions for future research and does not describe any limitations of this research project.

Author Response

(1). The topic of the research still remains the same "Administrators and Student on E-Learning, the Benefits and Impacts of Proper Implementation in Nigeria"

The literature review section and introductions has been updated with recent literatures and trends.

Discussion and Conclusion: As stated that the discussion didn't cover more and the need for further researches with limitations has been updated and clearly state. The conclusion also has been updated to address the discoveries from the study.

The corrections to this effect has been highlighted in reds to show the changes and adjustments been made to the work as regards your review.

Thank you so much.

Round 2

Reviewer 1 Report

Content 
~~~~~~~~~~~~
Does not relate to the special issue focus of impact of Covid19 to digital competence of Educators. It has only partly relate to the special issue title, having the use of e-learning at universities in focus.

In this version of manuscript, there are still lecturers considered to be administrators of e-Learning systems, as well as Heads of departments and deans (table 1). It is usually not the case - under "administrators" there are technical personnel dealing with the network/computer hardware infastructure and e-Learning software development or instalation and maintainance, while the e-Learning system users are both teaching staff (entering educational material, providing tests and feedback on students activities within the e-Learning system) and students (using educational material and having tests in e-Learning system). 

Typing errors
~~~~~~~~~~~~~~~
Casing
--------------
Title - instead of Student, there should be Students
Abstract - instead of Quest (Upper case) there should be quest (small letters). Table 3. should have "Poor" instead of "poor".
Symbols usage
---------------
Inappropriate use of characters, such as ";", example in last paragraph of section 3 - "findings show that; the success" 

Research methodology
~~~~~~~~~~~~~~~~~~~~~
It has been stated in the research methodology/design that the research has been based on questionnaires (which collect attitudes). It is, then, very unusual to have the performance of students mentioned in Table 3, regarding research question 3. The relationship between the educational administrators and performance of students could not be easily stated, as in this manuscript. What are the metrics related to the students' performance? Is it based on their marks at exams? Performance in this manuscript is simply stated as "poor", "good", "very good", but these qualifications are not supported by the metrics-based quantitative foundation. What is the numeric average mark of passed exams related to the qualification "poor" or "good"...?

Table 4. is related to research question 4, related to impact of the e-learning system usage - both on receiver (probably students) and provider (probably teaching staff + administrators). Items in table 4. are too wide (not precise and measurement-based), such as level of success, optimal results, cost of e-learning. Item 3 and 4 should be related, since the rate in examination malpractices should be taken as a measurable basis for comparison of e-Leaning and face to face mode of education.

Conclusion
~~~~~~~~~~~~
Conclusion still provides well-known sentences of benefits of e-learning, with insufficient level of emphasizing the significance of this research. 
There is a sentence which is highly unacceptable having "lives of both students and administrators has been tested". There is no life testing there in the paper, only attitudes about using e-learning and the effectiveness of it.
There are still no no future research directions.

Author Response

The edited and corrected version of the work are attached below with the sections corrected been highlighted with red colour.

the introduction also recent studies of e-learning concerning the Covid-19 has been updated and been included in the study.

The inclusion of lecturers as administrators is common in the Nigeria's educational system as most of them are seen servicing in different capacities and also helping in fixing problems that may arise in cause of e-learning implementation.

The research design has been optimized with the Research design, Instrumentation, Questionnaire Administration procedures etc.

The research section has also been modified and appropriate figures been assigned to 

The conclusion and area for further research has been optimized also with all highlighted in red.

Reviewer 2 Report

It will be helpful to include in the point of the methodology a flowchart that describes in detail the research design that was followed.

Author Response

All relevant corrections has been made from the introduction, the methodology, Research Design and Results has been clearly stated.

The conclusion and need for further research has been modified also.

The Flowchart was not used in the research initially because of the research pattern used, hence not needed and included.

Also the research design, and method of data collection method was with a questionnaire and not based on reviewed literatures, that will make provision for inclusion and exclusion criteria to necessitate flow chart.

The designs has been optimized and adjusted for clearer understanding of the design and method used in the study.

Attached below is the corrected file, and in highlighted colour below.

Round 3

Reviewer 1 Report

Typing errors

There are still many places when wrong capitalization has been done, such as within research question 3 (last word Learning should be lower case),  table 3 caption as well as at many other places, such as in "From table 4 above The impacts which", "Frequent reference to online resources for
Effective E-learning and optimum performance"...etc. It is innacceptable to have text of this manuscript published in this status.

Words selection

There are sometimes words that are inappropriately used, such as in "Contribution in knowledge varsity and self-development of e-learning compared with face-face learning." The word varsity is totally inadequate in this context. Another example -"The need for further research to checkmate..." - checkmate is term from chess playing ?!!!

Sometimes words are duplicated in a single sentence, example: "The need for further research to checkmate the hindrances of e-learning in Nigeria’s educational system is needed."

Punctuation and sentences formation

Missing commas in sentences, when needed, such as in the last paragraph of page 16. Sometimes sentences are too long and, therefore, could not be understood properly.

Abstract

Too long, with too much details with particular data i.e. numbers and percentages.

Research Methodology

It has been required to provide a more detailed research methods regarding the students' performance measurements basis. Research question and table 3. is related to students performance, but in the research methodology section there is not a single sentence stating the methods and sources for the performance data. This change is essential, since there is a core question - how e-Learning make impact on performance of students. So far, there is no correlation statistics made to prove statements regarding the e-Learning impact to their performance. There is not a single sentence describing the concept of students' performance and how is it measured and evaluated in this paper. All the paper is related to oppinions in questionnaire and, therefore, performance could not be mentioned in that context. The term "students performance" should be defined with more precise terms (such as average marks or something similar).  Tables 3 and 4 present simplified "results" about the e-Learning performance impact, while they are only presenting attitudes of students and administrators.

Conclusion section

Still consists of well-known sentences. Recommentation for further study are addressing the issues that have been addressed in this manuscript and it has only been focused on Nigeria, while future research directions should have a broad scientific contribution-related, no-country specific focus.

General conclusion about the manuscript

Even there are significant efforts made in manuscript writing and correction, there are still very serious flaws - no scientific contribution, many writing errors, research methodology based on questionnaire, while addressing performance and research questions and statements regarding students performance influenced by e-Learning usage (which requires correlation statistics analysis that would prove performance impact). 

Author Response

The abstract has been adjusted and reduced and those which are not relevant has been removed to a total of 260 words.

The methodology which shows criteria for grading has been updated.

The typing errors has been modified and corrected, with the article been reviewed  by the English department of MDPI.

The conclusion and further reading also has been modified.

Concerning the correlation of the impacts between students and administrators, correlation was not used initial in the work so adding that will be a new work all together, but the required modifications has been done.

Thank you.

Below is the corrected version.
